# Resource and location sharing in wireless networks

**Clara Stegehuis, Lotte Weedage**[ID] *

University of Twente, Enschede, The Netherlands

* l.weedage@utwente.nl

## Abstract

With more and more demand from devices to use wireless communication networks, there has been an increased interest in resource sharing among operators, to give a better link quality. However, in the analysis of the benefits of resource sharing among these operators, the important factor of co-location is often overlooked. Indeed, often in wireless communication networks, different operators co-locate: they place their base stations at the same locations due to cost efficiency. We therefore use stochastic geometry to investigate the effect of co-location on the benefits of resource sharing. We develop an intricate relation between the co-location factor and the optimal radius to operate the network, which shows that indeed co-location is an important factor to take into account. We also investigate the limiting behavior of the expected gains of sharing, and find that for unequal operators, sharing may not always be beneficial when taking co-location into account.

**Data Availability Statement:** The data underlying the results presented in the study are available from the Antenneregister of the Dutch Authority for Digital Infrastructure at https://antenneregister.nl/.

**Funding:** C.S. is supported through the Dutch Research Council (NWO) Veni grant 202.001. The

## Introduction

*Resource sharing* is a tool in wireless communications to enhance internet connectivity and capacity. Subscribers of one mobile network operator can then also use the available resources, such as cell towers and spectrum, of the other operators. This is in contrast with the standard regime where users can only connect to cell towers of their own operator, even when a different operator has a tower that would provide better quality of service. By resource sharing techniques, operators increase the redundancy of resources, which results in higher coverage (probability of having a connection) and fewer link failures for users [1, 2]. In this paper, we focus on sharing capacity, spectrum and base stations between different operators. Thus, subscribers of one mobile network operator will get access to the entire network of another operator [3]. We aim to quantify the benefits of network sharing over operators.

In [4], we have shown that resource sharing always brings benefits in terms of coverage and capacity (internet quality or *throughput*). Using location data of all base stations (BSs) in the Netherlands and population data of Statistics Netherlands (CBS), we showed through extensive simulations that having all operators work together as a single operator is beneficial. In this paper, we focus on a more different question: can we quantify the benefits of resource sharing mathematically?.

funders had no role in study design, data collection and analysis, decision to publish, or preparation of the manuscript.

**Competing interests:** The authors have declared that no competing interests exist.

Due to the increasing demand and new technologies, operators constantly need to deploy new BSs and decide where to place them. As building new cell towers on which BSs have to be placed can be very costly, operators often choose to build their BSs on existing cell towers with other BSs from different operators, which is called *co-location* of BSs (Fig 1). Indeed, in many national wireless networks several operators are often located closely to each other or have base stations on the same cell tower or building [5]. Co-location reduces the costs of the network, but also impacts the benefits of resource sharing in terms of less coverage, as has experimentally been shown in [6]. However, in mathematical analyses of the benefits of resource sharing, co-location is often overlooked, due to the complex expressions that one often obtains when looking at the standard measure for link quality: the Shannon capacity [7].

In this paper, we therefore analyse the performance of resource sharing when BSs of different operators are co-located, using tools from stochastic geometry. We model users and BSs by independent Poisson Point Processes (PPP) and connect users to BS according to the Poisson Boolean model [8]. Then, we provide a framework for co-location based on the *balls-into-bins* problem, where operator can choose to place a fraction $p$ of their BSs on an existing cell tower. We define a utility function that we call the *average user strength* and is based on the distance to a BS and the available resources at this BS. Maximizing utility function gives an optimal radius $r$ (Fig 2) that results in a *proportional fair* user association, which is a trade-off between the number of users that is disconnected and the resulting signal strength of a link.

We detect the threshold co-location factor $p$ for which sharing is beneficial for all operators. We then provide a small case study based on BS locations in the Netherlands that shows that our mathematical results can also be applied to real-world data.

In summary, we provide four main contributions. Firstly, using stochastic geometry, we derive an intricate relationship between the co-location probability $p$ and the optimal radius to operate the network, which shows the importance of taking co-location into account under resource sharing: the lower $p$, the lower the optimal radius $r$. Especially when the number of operators $N$ grows, this difference becomes more pronounced. Secondly, our analysis shows that when operators have unequal number of BSs there are two regimes, which strongly depend on the co-location parameter $p$. When $p$ is sufficiently small, sharing is beneficial for all operators. However, for a larger value of $p$, sharing is not beneficial for the larger operator anymore. We identify a lower bound of this transition point, which shows that even in the

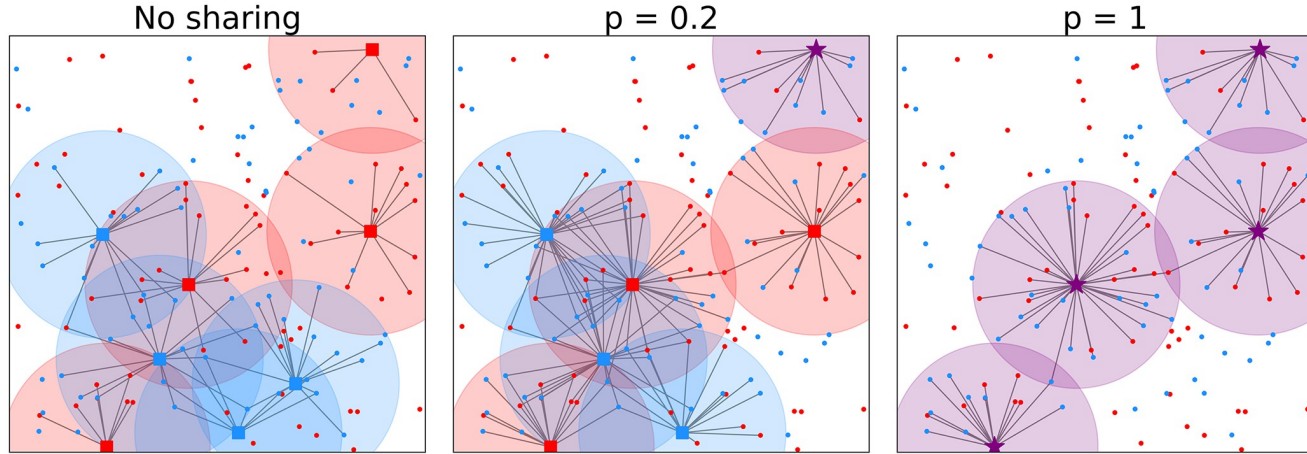

**Fig 1. Every users connects to BSs within radius *r*.** Example of a setting with no sharing and settings with sharing (every user connects to its own BS) and a setting with sharing and two different co-location factors *p*. There are BSs and users of two providers, denoted by the red an blue squares. Co-located BSs are denoted by a purple star.

(a)                                    (b)

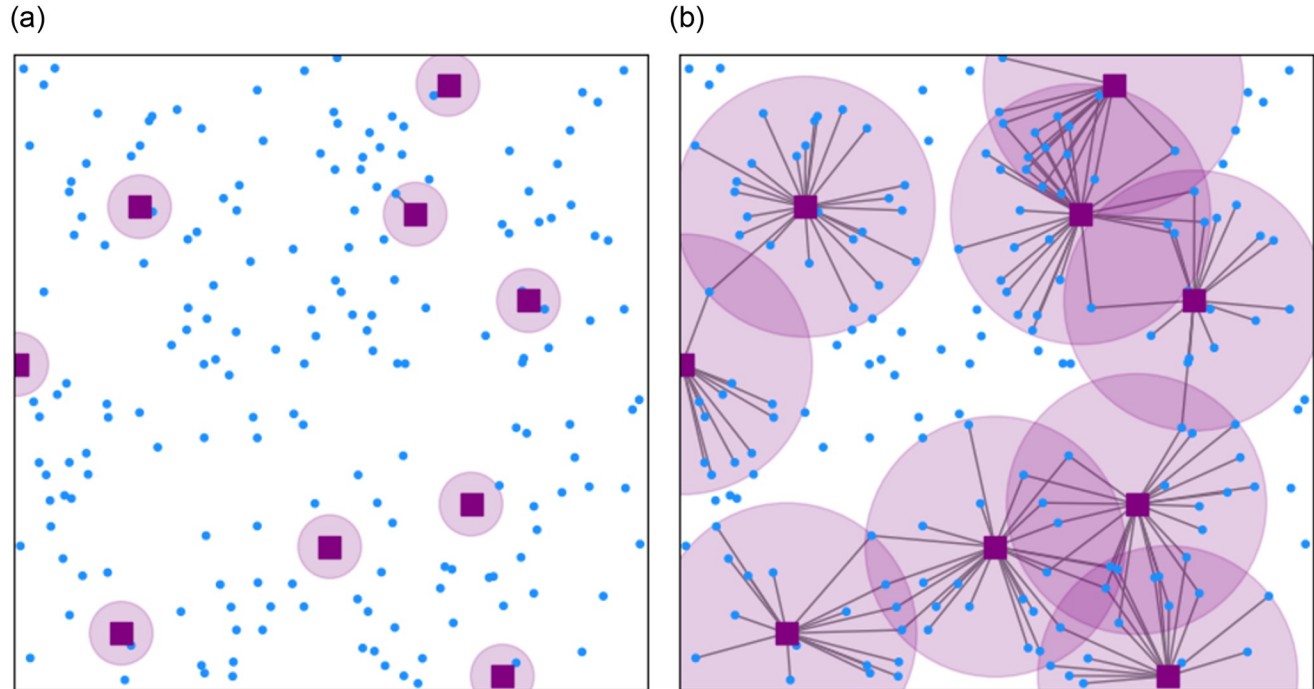

**Fig 2. Network under different values of *r*.** Red squares are base stations and the blue dots are users. The red circle denotes the region in which users connect to that specific base station. (a) $r = 75$. (b) $r = 308$.

limit when the ratio between the class sizes tends to zero, there is still a regime of *p* where sharing is beneficial for both parties. Furthermore, for a large number of operators *N*, we show that the gain of sharing scales as $N^{1/3}$. Finally, we perform a case-study on real-world data which shows that even in a non-homogeneous spatial setting, our theoretical results apply.

## Literature

Mathematical literature on communication systems often focuses on unipartite graphs. In these graphs, users communicate with other users that are within a certain range [9], or it models the backbone network between BSs [10]. Often, these works establish conditions for a giant component or a percolation threshold to exist [11], as a giant component in this model means that most users are able to communicate. However, in a realistic setting, users do not communicate through other users' phones, but through BSs. This means that to investigate the quality of service in a wireless network, unipartite graphs do not suffice. Therefore, we model the wireless network as a bipartite graph model with two types of nodes: users and BSs, where an edge can only exist between a user and a BSs, similarly to [9].

Two types of resources can be shared in wireless networks: *infrastructure* and *spectrum*. Under spectrum sharing, frequency bands of operators are re-used by other operators, which increases the diversity of frequency bands that a BS can use. In this research, we focus on infrastructure sharing, where operators can place their BSs at the same location to reduce infrastructure costs [12]. Several works on the techniques and algorithms for resource sharing exist. The general conclusion is that sharing benefits the network in terms of throughput and coverage based on simulations [4].

In [13] the authors give a framework for optimal resource sharing based on game theory, where operators can decide whether to work together or not. Such game theoretic approaches have also been performed by [2]. However, this line of research focuses on *how* to allocate available resources such as spectrum and BSs in a network in which BS locations are already fixed. In this paper, we take a step back and look at the design and geometry of the network and analyse when sharing is beneficial, assuming all resources are shared.

In terms of the benefits of infrastructure and spectrum sharing [14], provides an analysis of the probability that a user can connect to any BS (*coverage probability*) under spectrum sharing with co-location. They show that under sharing, operators need less resources to provide the same service level. However, they do not provide analyses on gains and user benefits. In [15] the authors show a trade-off between coverage and data rate under spectrum, infrastructure and full sharing. Moreover, they propose a clustered Poisson Point process to model a 2-operator network, where BSs of different operators are located close to each other. They show that spatial diversity is an important factor in the coverage probability and that these three methods of sharing result in different benefits to the users (coverage and/or data rate gain). That spatial diversity is important is also shown experimentally in [6]. Moreover, when the operators have different BS densities they will not benefit equally from sharing [16]. However, both papers do not analyse the gain under different co-location factors. A similar result is given in [17], who propose a model with co-location of BSs based on real-world data and identify that approximately 40% of BSs are co-located. However, there are no clear results on the improvement of sharing under co-location, which we will provide in this paper.

## Model

The setting is as follows. In total there are $N$ operators, which we call *types*. Every type $k \in \{1, \ldots, N\}$ contains users $i_k \in U_k$ and base stations $j_k \in \mathcal{B}_k$, which are independently and identically distributed as Poisson Point Processes (PPP) with densities $\lambda_U^{(k)}$ and $\lambda_{BS}^{(k)}$, respectively. An edge between user $i_k$ and base station $j_k$ exists when $d(i_k, j_k) \leq r$, where $d(\cdot, \cdot)$ denotes the distance between two points. This model is similar to the Poisson Boolean model [8] or disk model (Fig 1). The above-mentioned model gives $N$ separate networks. To show the benefits of sharing, we combine these networks in terms of resources and location.

In the case of *infrastructure sharing* every user can use any BS, no matter the type. In the above-mentioned model, this means that edges can also exist between $i_k$ and $j_l$ for $j \neq l$ if $d(i_k, j_l) \leq r$. For *location sharing*, BSs of different types can be placed on the same location to reduce costs for building new towers. We assume that BS locations of type $k = 1$ are fixed and the BSs of types $k > 1$ can share their location with BSs of type 1. The co-location factor $p$ indicates which percentage of the BSs of types $k > 1$ are co-located with BSs of type 1. Thus, if $p$ is equal to 1 all BSs of different types $k > 1$ are located on the same cell tower as BSs of type 1 and $p = 0.5$ means that half of the BSs of types $k > 1$ will be co-located with the BSs of type 1 (Fig 1). With $\mathcal{T}$ we define the set of all cell tower locations.

We assume that the type 1 always has the largest density: $\lambda_{BS}^{(1)} \geq \lambda_{BS}^{(k)}$ and $\lambda_U^{(1)} \geq \lambda_U^{(k)}$ for type $k > 1$. All BS and user densities for $k > 1$ can then be described in terms of $\lambda_{BS}^{(1)} = \lambda_{BS}$:

$$\lambda_{BS}^{(k)} := \beta_k \lambda_{BS}, \text{ and } \lambda_U^{(k)} := \beta_k \lambda_U, \tag{1}$$

where $0 < \beta_k < 1$ is the fraction $\lambda_{BS}^{(k)}/\lambda_{BS}^{(1)}$, assuming the user-to-BS-ratio is the same for every type. Moreover, since the benefits of sharing resources are largest in a dense network, we assume that the user density is much larger than the BS density. Thus, together with the co-location model as described above, we obtain the following cell tower and user density of the

entire network:

$$\tilde{\lambda}_T = \left(1 + (1-p)\sum_{k=2}^{N} \beta_k\right)\lambda_{\text{BS}}, \tag{2}$$

$$\tilde{\lambda}_U = \left(1 + \sum_{k=2}^{N} \beta_k\right)\lambda_{\text{U}}. \tag{3}$$

The resulting network exists of users $\mathcal{U} = \bigcup_{k=1}^{N} \mathcal{U}_k$ and towers $\mathcal{T} = \bigcup_{k=1}^{N} \mathcal{B}_k$ are distributed by PPPs with densities $\tilde{\lambda}_U$ and $\tilde{\lambda}_T$, respectively.

## Link strength analysis

A typical measure to measure the quality of a connection between user $i$ and BS $j$ is the Shannon channel capacity $c$, which is a function of the bandwidth (*resources*) and the *signal-to-noise-ratio* (SNR):

$$c_{ij} = \frac{wC_j}{D_j}\log\left(1 + K \cdot R_{ij}^{-\alpha}\right). \tag{4}$$

In this equation, $w$ is the bandwidth (*resources*) per BS, $C_j$ the number of BSs that are co-located on cell tower $j$ and $R_{ij}$ is the distance between user $i$ and tower $j$ and $D_j$ is the degree of BS $j$, which we define as the number of users that is connected to BS $j$. Moreover, $K$ is equal to the transmission power of a BS divided by the background noise due to other signals. Thus, $K \cdot R_{ij}^{-\alpha}$ represents the SNR, which is a function that decreases over distance with exponent $\alpha$ indicating how fast the signal decays (*free space path loss*, [18]). In wireless networks, next to background noise the signal is affected by interference of signals from other BSs. We assume in our network model that the interference is negligible due to for example neighboring BSs operating on different frequencies or due to mmWave BSs operating with non-interfering beams [19].

In our network model, users connect to all BSs that are within distance $r$. When $r$ is large users connect on average to many BSs while for $r$ small many users might not connect to a BS at all. Therefore, the choice of $r$ should result in a network with few disconnected users while keeping the quality of every link high.

The channel capacity as defined in (4) is a decreasing function in $r$ from the point where every user is connected to at least one BS. If $r$ is too small, not every user will be connected which will result in a lower average channel capacity. Then, the typical degree $D_j$ increases for larger $r$ since more users will connect to every BS and users can be further away from the BS, so typically $R_{ij}$ increases as well. Thus, the optimal $r$ in this case is close to 0 (Fig 3) where on average every BS only serves its closest user [20]. In summary, optimal channel capacity prefers serving only a few close by users with a large rate over serving more (and further away) users with a lower rate.

Therefore, we do not use the channel capacity to find the optimal radius $r$, but we define a logarithmic utility function related to the channel capacity. Such a logarithmic utility function is common in telecommunications to ensure a (proportional) fair network [20, 21]. In this utility function, which we call the *user strength*, we give priority to users further away and do not give any more resources to the well-served users close to BSs.

Similar to the channel capacity, the user strength $S_i$ depends on the degree of a tower, the distance to this tower and the number of resources available at that tower. However, instead of multiplying the resources with the logarithm of the SNR, we take the logarithm of all random

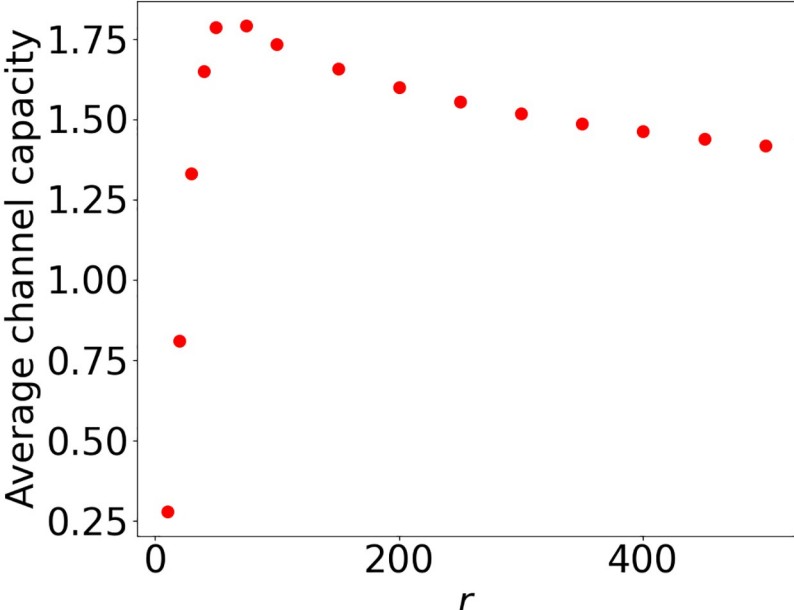

**Fig 3. Average channel capacity per user for different values of $r$ with $K = 111$ dBm, $\alpha = 2$ and $w = 10$ MHz.**

variables for analytical tractability and to prioritize all random variables equally. When the SNR is small, this utility function is similar to taking the logarithm of the channel capacity by the first-order Taylor expansion.

$$S_i = \sum_{j \in \mathcal{T}^i} \log\left(\frac{wC_j}{D_j R_{ij}}\right), \tag{5}$$

where $w$ is the bandwidth (*resources*) per BS, $C_j$ the number of BSs that are co-located on cell tower $j$ and $R_{ij}$ is the distance between user $i$ and tower $j$ and $D_j$ is the degree of BS $j$. The set $\mathcal{T}^i$ is the set of all towers that are within radius $r$ of user $i$. In the rest of the paper, we assume $\alpha = 1$ but all analyses will hold for other values of $\alpha$ as well.

For the setting in Fig 3 we show with simulations that $r \approx 75$ maximizes the total channel capacity per user. This radius will result in 99% of the users being disconnected (Fig 2a). Maximizing our proposed utility as defined in (5) results in an optimal radius of $r = 308$ (Fig 2b), with only 30% of the users disconnected.

In the following sections, we derive the expected strength and the regime for which this strength is maximal. Moreover, we provide an analysis of the benefit of sharing for different types and in case of unequal BS densities.

**Theorem 1** (Expected strength). *For $\tilde{\lambda}_U \pi r^2 > 1$:*

$$\mathbb{E}[S] = \tilde{\lambda}_T \pi r^2 \left( \log(w) + \mathbb{E}[\log(C)] - \log\left(\tilde{\lambda}_U \pi r^3\right) + \frac{1}{2} \right)\left( 1 + O\left(\frac{1}{\tilde{\lambda}_U \pi r^2}\right)\right). \tag{6}$$

*For large N and $\beta_k = 1$ for all $k > 1$:*

$$\mathbb{E}[\log(C)] = \frac{1}{1 + (1-p)(N-1)}\left( \log(1 + p(N-1)) - \frac{p(1-p)(N-1)}{(1 + p(N-1))^2} + O\left(\frac{1}{N^2}\right)\right) \tag{7}$$

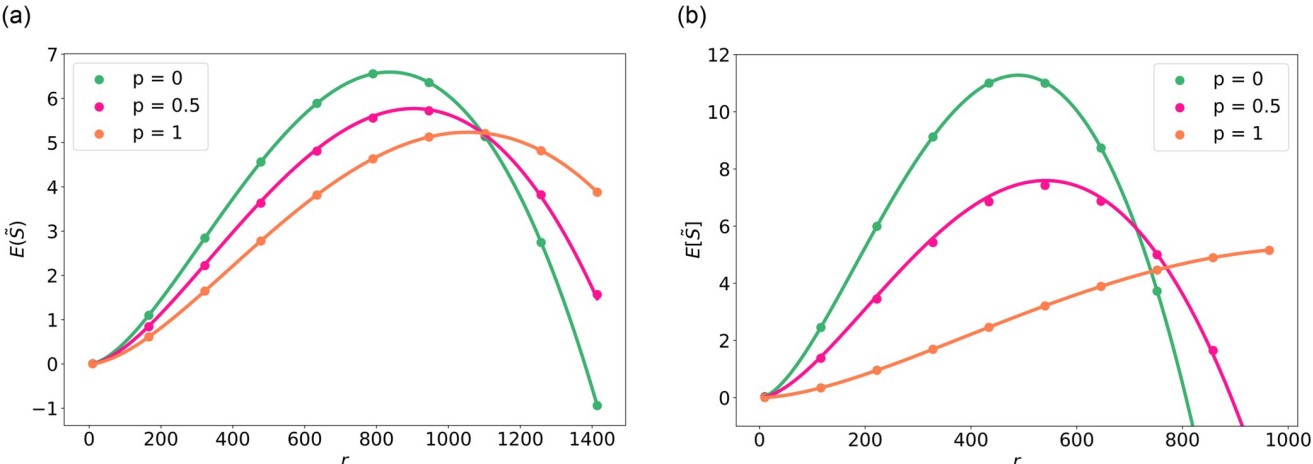

**Fig 4. Simulation (markers) and the approximation (Eq (9), denoted by the solid lines).** (a) $N = 2$. (b) $N = 10$.

*Moreover, when $N = 2$, for all $\beta_2 \leq 1$:*

$$\mathbb{E}[\log(C)|N = 2] = \frac{\beta_2 p \log(2)}{1 + (1 - p)\beta_2}. \tag{8}$$

The proof of this theorem can be found in the section Proof of Theorem 1.

In the rest of this paper, we assume $\tilde{\lambda}_U \pi r^2$ will be large and use the approximated expected strength $\mathbb{E}[\tilde{S}]$:

$$\mathbb{E}[\tilde{S}] = \tilde{\lambda}_T \pi r^2 \left( \log(w) + \mathbb{E}[\log(C)] - \log\left(\tilde{\lambda}_U \pi r^3\right) + \frac{1}{2} \right). \tag{9}$$

Fig 4 shows the expected strength from simulations of the model as described in the section Model for in total 100 iterations on an area of $1000 \times 1000$m. In all simulations, we use $w = 10 \cdot 10^6$, $\lambda_U = 1 \cdot 10^{-3}$ and $\lambda_{BS} = 1 \cdot 10^{-6}$, unless stated otherwise. This figure shows that the approximation given in Theorem 1 performs well compared to the outcomes of the simulations for large $N$ as well as for $N = 2$. Fig 4 suggests that there is a unique value of $r$ for which the strength is maximal, which depends on the co-location factor. The following theorem therefore provides this optimal radius and its associated strength:

**Theorem 2** (Optimal radius and strength). *For the expected strength given in Theorem 1, the optimal radius $r$ equals*:

$$r^{opt} = \sqrt[3]{\frac{w}{\tilde{\lambda}_U \pi} e^{-1 + \mathbb{E}[\log(C)]}}, \tag{10}$$

*which gives the following optimal strength*:

$$\mathbb{E}(S^{opt}) = \frac{3e^{-2/3} \tilde{\lambda}_T}{2} \sqrt[3]{\frac{e^{2\mathbb{E}[\log(C)]} \pi w^2}{\tilde{\lambda}_U^2}}, \tag{11}$$

*with* $\mathbb{E}[\log(C)]$ *as given in* (7). *Moreover, for* $N \to \infty$,

$$\sqrt[3]{N}r^{opt} \xrightarrow[N \to \infty]{} \sqrt[3]{\frac{w}{\lambda_U \pi e}}, \tag{12}$$

$$\frac{\mathbb{E}[S^{opt}]}{\sqrt[3]{N}} \xrightarrow[N \to \infty]{} 1 - p. \tag{13}$$

*Proof.* To find $r^{\mathrm{opt}}$, we take the derivative of (9) and set this derivative equal to 0:

$$\frac{\mathrm{d}\mathbb{E}[S]}{\mathrm{d}r} = 2\tilde{\lambda}_\mathrm{T}\pi r(\log(W) + \mathbb{E}[\log(C)] - \log(\tilde{\lambda}_\mathrm{U}\pi r^3) - 1) = 0, \tag{14}$$

$$r^{opt} = \sqrt[3]{\frac{w}{\tilde{\lambda}_\mathrm{U}\pi}e^{-1+\mathbb{E}[\log(C)]}}, \tag{15}$$

where $\mathbb{E}[\log(C)]$ is defined in Theorem 1. Then, we fill in $r^{\mathrm{opt}}$ in (6) to obtain $\mathbb{E}(S^{opt})$, resulting in (11).

Since $\mathbb{E}[\log(C)] \to 0$ when $N \to \infty$, we obtain (12), which shows that $r^{opt}$ scales with $1/\sqrt[3]{3}$. Moreover, in the definitions of $\tilde{\lambda}_\mathrm{T}$ and $\tilde{\lambda}_\mathrm{U}$ as given in (2) and (3), the sum over $\beta_k$ simplifies to $(N-1)$, resulting in the scaling limit in (13).

For $p = 1$, the optimal radius $r^{opt} = \sqrt[3]{\frac{w}{\pi\lambda_U e}}$ does not depend $N$. In this regime, all BSs of types $k > 1$ are co-located and the increase in number of users and number of resources per cell tower cancel out for $p = 1$.

## Sharing gain

We now investigate the benefit of sharing given the optimal radius and strength in Theorem 2 for two cases: (1) $N = 2$ and $\beta_2 \leq 1$ and (2) large $N$ with $\beta_k = 1$ for all $k$.

**$N = 2$, different BS and user densities.** We consider $N = 2$ where type $k = 2$ has a fraction $\beta_2 \leq 1$ compared to the number of BSs of type 1. Fig 5 plots an example of the expected strength for type 1 and type 2 when they operate on their own and when they share resources. In the sharing scenario, co-location leads to a lower average strength compared to the no-

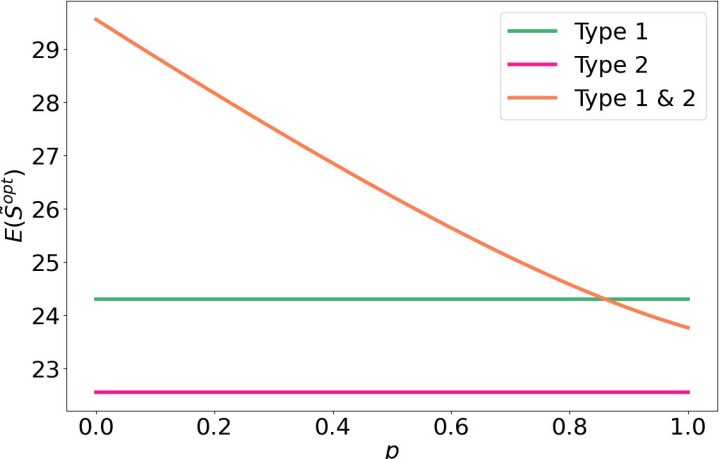

**Fig 5. Expected optimal strength for two types, with $\beta_2 = \frac{4}{5}$.**

sharing scenario for type 1 for some values of $p$, while sharing will always be beneficial to users of type 2. For type 1 only benefits from resource sharing, when $p$ is below approximately 0.8. The following corollary provides an upper bound on the co-location factor such that both operators benefit:

**Corollary 1** (Sharing gain $N = 2$). *For $N = 2$,*

$$G^{type1}(p) := \frac{\mathbb{E}[\tilde{S}^{opt}|N=2]}{\mathbb{E}[\tilde{S}^{type1}]} = (1 + (1-p)\beta_2)\sqrt[3]{\frac{2^{\frac{2\beta_2 p}{1+\beta_2(1-p)}}}{(1+\beta_2)^2}}, \tag{16}$$

$$G^{type2}(p) := \frac{\mathbb{E}[\tilde{S}^{opt}|N=2]}{\mathbb{E}[\tilde{S}^{type2}]} = \sqrt[3]{\frac{1}{\beta_2}}G^{type1}(p), \tag{17}$$

*where $G^{type1}$ is the gain of sharing for type 1 and $G^{type2}$ the gain of sharing for type 2. Type 2 will always benefit from sharing as $G^{type2}(p) \geq 1$ for all $p$ and $\beta_2 \in [0, 1]$. Moreover, when*

$$p \leq 1 - \frac{3(1 + (1+\beta_2)^{2/3}4^{-\beta_2/3})}{\beta_2((1+\beta_2)\log(4) - 3)}, \tag{18}$$

*both $G^{type1}(p)$ and $G^{type2}(p)$ will be greater than or equal to 1, which indicates that both types benefit from sharing.*

*Proof.* For $N = 2$, the optimal strength given in Theorem 2 is:

$$\mathbb{E}[\tilde{S}^{opt}|N=2] = \frac{3e^{-2/3}}{2}(1 + (1-p)\beta_2)\lambda_{BS}\sqrt[3]{\frac{2^{\frac{2\beta_2 p}{1+\beta_2(1-p)}}\pi w^2}{(1+\beta_2)^2\lambda_U^2}}. \tag{19}$$

To quantify the benefit of sharing, we compare the expected strength under sharing to the expected strength for both types separately when they will not share ($N = 1$). For type 1 the BS and user density is $\lambda_{BS}$ and $\lambda_U$ and for type 2 $\beta_2\lambda_{BS}$ and $\beta_2\lambda_U$, respectively:

$$\mathbb{E}[\tilde{S}^{type1}] = \frac{3e^{-2/3}}{2}\lambda_{BS}\sqrt[3]{\frac{\pi w^2}{\lambda_U^2}}, \tag{20}$$

$$\mathbb{E}[\tilde{S}^{type2}] = \frac{3e^{-2/3}}{2}\beta_2\lambda_{BS}\sqrt[3]{\frac{\pi w^2}{(\beta_2\lambda_U)^2}} = \sqrt[3]{\beta_2}\mathbb{E}[\tilde{S}^{type1}]. \tag{21}$$

The gains $G^{type1}(p)$ and $G^{type2}(p)$ can be found by filling in these equations.

Since $\beta_2 \leq 1$, the expected user strength for users of type 2 without sharing will be always smaller or equal to (20). Therefore, we find the regime of $p$ and $\beta_2$ such that $G^{type1}(p) \geq 1$. We take the first-order Taylor expansion of (20) around $p = 1$:

$$\begin{aligned} G^{type1}(p) &= (1 + (1-p)\beta_2)\sqrt[3]{\frac{2^{\frac{2\beta_2 p}{1+\beta_2(1-p)}}}{(1+\beta_2)^2}} \\ &\leq \frac{1}{3}\left(3 - \beta_2(1-p)(\log(4) - 3) - (\beta_2)^2(1-p)\log(4)\right). \end{aligned} \tag{22}$$

To find the regime of $p$ for which $G^{type1}(p)$ is greater than or equal to $p$, we solve:

$$\frac{1}{3}\left(3 - \beta_2(1-p)(\log(4) - 3) - (\beta_2)^2(1-p)\log(4)\right) \geq 1. \tag{23}$$

By solving this inequality, we obtain (18). Thus, for $p$ at least smaller or equal to (18) sharing is beneficial for both type 1 and type 2 users.

Moreover, we show that $G^{type2}(p)$ will always be larger than 1. The expected strength for $N = 2$ (19) decreases in $p$, since:

$$\frac{d\mathbb{E}[\tilde{S}^{opt}|N=2]}{dp} = \frac{\beta_2(-3 + \log(4) + \beta_2(-3(1-p) + \log(4)))}{3 + 3\beta_2(1-p)}\sqrt[3]{\frac{2^{\frac{2\beta_2 p}{1+\beta_2(1-p)}}}{(1+\beta_2)^2}}. \tag{24}$$

The third root and the denominator are always positive, and

$$-3 + \log(4) + \beta_2(-3(1-p) + \log(4)) < 0, \tag{25}$$

for all $\beta_2 \in [0, 1]$. Therefore, $\mathbb{E}[\tilde{S}^{opt}|N = 2]$ decreases in $p$. This means that we can $G^{type2}(p) \geq G^{type2}(1)$ for all $p \in [0, 1]$:

$$G^{type2}(1) = \sqrt[3]{\frac{4^{\beta_2}}{\beta_2(1+\beta_2)^2}} \geq 1, \tag{26}$$

for all $\beta_2 \in [0, 1]$. Thus, sharing for type 2 will always be beneficial.

Fig 6 shows that the approximation to the upper bound given in (18) is close to the real value, which we found numerically. Moreover, this figure shows that even for a very small $\beta_2$, the value of $p$ has to be smaller than 0.65 for sharing to be beneficial, which indicates that when less than a fraction of 0.65 is co-located, sharing is beneficial in all settings.

**Large N, equal BS and user densities.** Now, we provide results for the scenario with $N$ types of equal BS and user density. Fig 7 shows the expected strength given the optimal radius $r$ for different values of $p$ and $N$, compared to the optimal strength under no sharing. This figure shows that for a small co-location fraction $p$, sharing always results in a better average

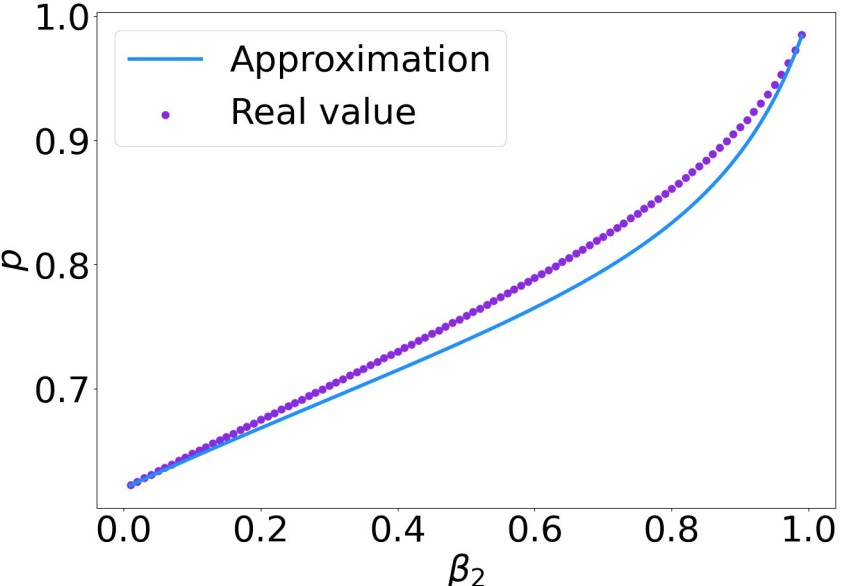

**Fig 6. Value of $p$ for which $G^{type1}(p) = 1$ and the upper bound given in (18).**

(a)

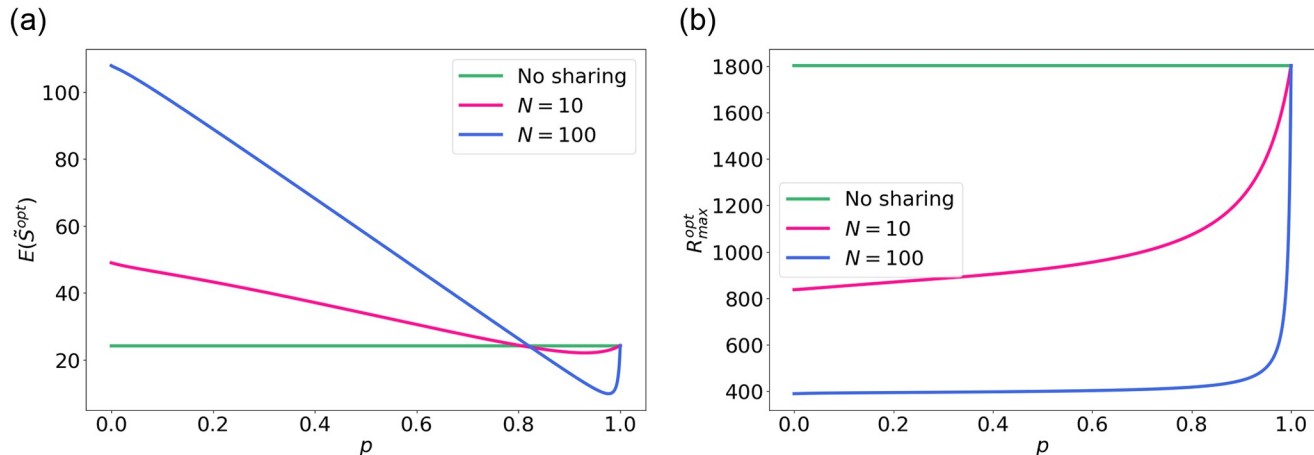

(b)

**Fig 7. Optimal strength and the corresponding values for $r^{opt}$.** (a) Expected strength for optimal radius $r^{opt}$. (b) Optimal radius $r^{opt}$.

strength. However, for larger $p$, sharing results in a lower user strength compared to $N = 1$. The decrease in strength has the following reason: for large $p$, the optimal radius $r$ increases (Fig 7b). When the radius increases, more users will be connected to each BS (Fig 8), which means that the resources available at each BS need to be shared among more users. Moreover, links at larges distances provide a lower strength, leading to a lower average strength.

The following provides an upper bound on the co-location factor such that all $N$ providers benefit from sharing.

**Corollary 2** (Sharing gain large $N$). *For $N$ large and $\beta_k = 1$ for all types $k$,*

$$G^N(p) = \frac{\mathbb{E}[\tilde{S}^{opt}]}{\mathbb{E}[\tilde{S}^{type1}]} = \frac{1 + (1-p)(N-1)}{\sqrt[3]{N^2}} e^{-\frac{2}{3}\mathbb{E}[\log(C)]}, \tag{27}$$

*where $G^N(p)$ is the gain of sharing $N$ types. Moreover, when*

$$p \leq \frac{N - \sqrt[3]{N^2}}{N - 1}, \tag{28}$$

*sharing will result in a higher average user strength compared to no sharing.*

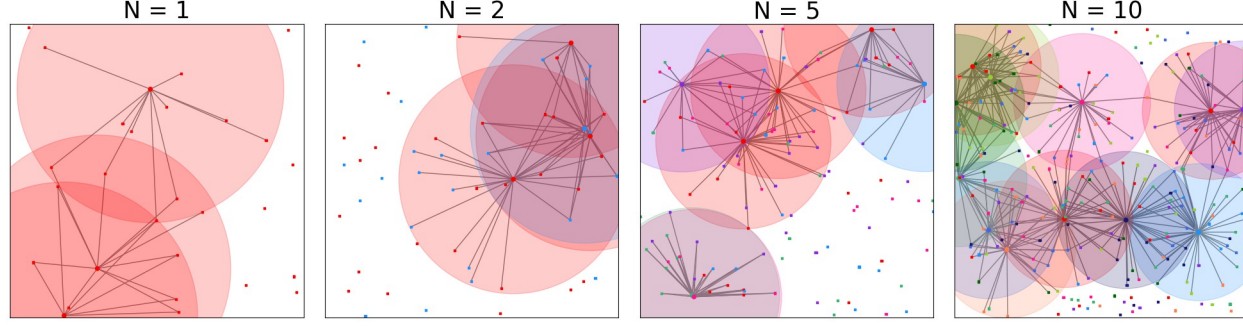

**Fig 8. Network with $p = 0.5$ and optimal radius $r^{opt}$, which changes for different values of $N$.**

*Proof.* We find the gain $G^N(p)$ by filling in (9) and (20) directly, resulting in (27). To find the regime of $p$ for which $G^N(p) \geq 1$, we again simplify (27) and use an upper bound for $\mathbb{E}[\log(C)]$:

$$e^{-\frac{2}{3}\mathbb{E}[\log(C)]} \leq 1. \tag{29}$$

Therefore,

$$G^N(p) \leq \frac{1 + (1-p)(N-1)}{\sqrt[3]{N^2}} = \tilde{G}^N(p). \tag{30}$$

Now, solving $\tilde{G}^N(p) \geq 1$ for $p$ results in (28).

Fig 9 shows the real value and approximation of the threshold value of $p$. Again, the approximation is close to the real upper bound when $N$ is large enough. Compared to Fig 6, the $p$ value for which sharing is not beneficial any more is higher, starting around $p = 0.8$.

## Proof of Theorem 1

To prove Theorem 1, we first derive the probability distributions of the number of resources $C$ and the cell tower degree $D_T$ in the following two lemmas:

**Lemma 1** (Expectation of the number of resources at every tower). *When $N$ is large and $\beta_k = 1$ for all $k > 1$:*

$$\mathbb{E}[\log(C)] = \frac{1}{1 + (1-p)(N-1)} \left( \log(1 + p(N-1)) - \frac{p(1-p)(N-1)}{(1 + p(N-1))^2} + O\left(\frac{1}{N^2}\right) \right) \tag{31}$$

*Moreover, when $N = 2$ and $N = 3$, for all $\beta_2$ and $\beta_3$, not necessarily equal to* 1:

$$\mathbb{E}[\log(C)|N = 2] = \frac{\beta_2 p \log(2)}{1 + (1-p)\beta_2}. \tag{32}$$

*Proof.* In the proposed model, BSs of type 1 by definition have their own cell tower while BSs

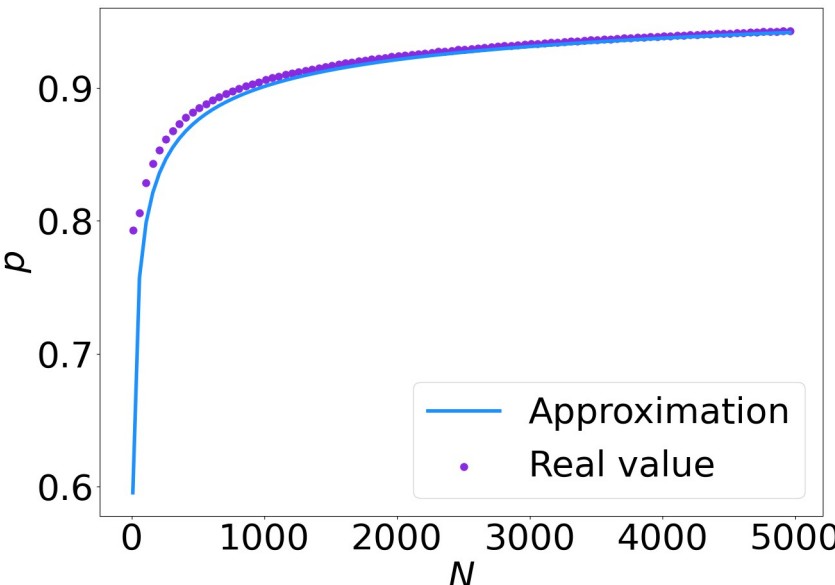

**Fig 9. Value of $p$ for which $\tilde{G}^N(p) = 1$ and the upper bound given in (28).**

of types $k \neq 1$ can either be on their own cell tower, or co-located on a cell tower of type 1. Thus, we distinguish two different classes of cell towers: towers of type 1 and towers of type $k \neq 1$. The number of resources (BSs) per cell tower $C$ is equal to 1 for the second class and $C \geq 1$ for the first class. We obtain the following probability distribution for $C$:

$$\mathbb{P}(C = 1, k \neq 1) = \mathbb{P}(k \neq 1) = \frac{(1-p)\sum_{k=2}^{N}\beta_k}{1 + (1-p)\sum_{k=2}^{N}\beta_k} =: s, \tag{33}$$

$$\mathbb{P}(C = i, k = 1) = \mathbb{P}(k = 1)\mathbb{P}(C = i|k = 1) = (1-s)\mathbb{P}(C^{(1)} = i - 1), \tag{34}$$

where $C^{(1)}$ follows a Poisson binomial distribution with parameters $\{p\beta_k\}_{k\in[2,N]}$. When $\beta_k = 1$ for all $k > 1$, the probability distribution of $C^{(1)}$ simplifies to a binomial distribution with $N-1$ trials and probability $p$. Using the definition of expectation and the probability function given in (33) and (34), we obtain:

$$\begin{aligned}
\mathbb{E}[\log(C)] &= \sum_{i=1}^{\infty} \log(i)\mathbb{P}(C = i) \\
&= \log(1)\mathbb{P}(C = 1, k \neq 1) + \sum_{i=1} \log(i)\mathbb{P}(C = i, k \neq 1) \\
&= (1-s)\sum_{i=0}^{\infty} \log(i+1)\mathbb{P}(C^{(1)} = i) \\
&= (1-s)\mathbb{E}[\log(C^{(1)} + 1],
\end{aligned} \tag{35}$$

with $s$ defined in (33). We distinguish two cases: (1) $N = 2$ with unequal BS and user density and (2) $N$ large with equal user density (i.e. $\beta_k = 1$). In the first case, we find $\mathbb{E}[\log(C)]$ directly using the definition of expectation, resulting in (32).

As there is no closed-form analytic expression of the expectation of the logarithm of a binomial random variable plus one, we use a first-order Taylor expansion for the second case:

$$\begin{aligned}
\mathbb{E}[\log(C^{(1)} + 1)] &= \log(\mathbb{E}[C^{(1)} + 1]) - \frac{\mathrm{Var}(C^{(1)})}{2(1 + \mathbb{E}[C^{(1)}])^2} + O\left(\frac{\mathbb{E}[(C^{(1)} - \mathbb{E}[C^{(1)}])^3]}{(1 + \mathbb{E}[C^{(1)}])^3}\right) \\
&= \log(1 + p(N-1)) - \frac{p(1-p)(N-1)}{(1 + p(N-1))^2} + O\left(\frac{p(1 - 3p + 2p^2)(N-1)}{(1 + p(N-1))^3}\right).
\end{aligned}$$

Plugging this result for $\mathbb{E}[\log(C^{(1)} + 1)]$ into (35), $\mathbb{E}[\log(C)]$ results in (34).

**Lemma 2** (Tower degree). *For $\tilde{\lambda}_U \pi r^2$ large*:

$$\mathbb{E}[\log(\tilde{D}_T)] = \log\left(1 + \tilde{\lambda}_U \pi r^2\right) - \frac{\tilde{\lambda}_U \pi r^2}{2(1 + \tilde{\lambda}_U \pi r^2)^2} + O\left(\frac{1}{(\tilde{\lambda}_U \pi r^2)^2}\right), \tag{36}$$

*where $\tilde{D}_T$ is the size biased degree of a cell tower* [22].

*Proof.* The degree distribution of a cell tower is equal to the number of users in a circle with radius $r$ around that tower that can connect to this tower. Given that users are distributed by a PPP,

$$p_n := \mathbb{P}(D_T = n) = \mathbb{P}(N_U(\pi r^2) = n) = \frac{(\tilde{\lambda}_U \pi r^2)^n}{n!} e^{-\tilde{\lambda}_U \pi r^2}, \tag{37}$$

$$\mu := \mathbb{E}[D_T] = \tilde{\lambda}_U \pi r^2. \tag{38}$$

Moreover, the expectation of the size biased degree distribution $\tilde{D}_T$ is:

$$\mathbb{E}[\tilde{D}_T] = \sum_{n=0}^{\infty} n \cdot \frac{np_n}{\mathbb{E}[D_T]} = 1 + \tilde{\lambda}_U \pi r^2. \tag{39}$$

Similar to the proof in Lemma 1, we use a first-order Taylor expansion around $\mathbb{E}[\tilde{D}_T]$:

$$\begin{aligned}
\mathbb{E}[\log(\tilde{D}_T)] &= \log(1 + \mu) - \frac{\text{Var}(\tilde{D}_{BS})}{2(1 + \mu)^2} + O\left(\frac{\mathbb{E}((\tilde{D}_{BS} - (1 + \mu))^3)}{6(1 + \mu)^3}\right) \\
&= \log\left(1 + \tilde{\lambda}_U \pi r^2\right) - \frac{\tilde{\lambda}_U \pi r^2}{2(1 + \tilde{\lambda}_U \pi r^2)^2} + O\left(\frac{1}{(\tilde{\lambda}_U \pi r^2)^2}\right).
\end{aligned} \tag{40}$$

*Proof.* (*Proof of Theorem 1.*) The strength defined in (5) consists of the independent random variables $C$, $\tilde{D}_T$ and $R$. We removed the indices as all these random variables do not depend on the user $i$ nor the BS $j$. Therefore, we obtain:

$$\begin{aligned}
\mathbb{E}[S] &= \mathbb{E}[D_U]\mathbb{E}\left[\log\left(\frac{wC}{\tilde{D}_T R}\right)\right] \\
&= \mathbb{E}[D_U](\log(w) + \mathbb{E}[\log(C)] - \mathbb{E}[\log(\tilde{D}_T)] - \mathbb{E}[\log(R)]).
\end{aligned} \tag{41}$$

Moreover, $\tilde{D}$ is the size-biased degree of $D$, which is the degree of a BS given that a user connects to this BS. In Lemmas 1 and 2 we provide the expectations of $\log(C)$ and $\log(\tilde{D}_T)$. The expected degree of a user is equal to $\tilde{\lambda}_T \pi r^2$, as a user will connect to all BSs within radius $r$ and BSs are distributed by a PPP. Lastly, the distance $R$ between a user and a cell tower has the following probability distribution:

$$f_R(x) = \begin{cases} \dfrac{2x}{r^2}, & \text{if } 0 < x \leq r, \\[2mm] 0 & \text{else.} \end{cases} \tag{42}$$

The expectation of $\log(R)$ then follows:

$$\mathbb{E}[\log(R)] = \int_0^r \log(x) \frac{2x}{r^2} \, dx = \log(r) - \frac{1}{2}. \tag{43}$$

Putting all expectations together, we get the following expression for $\mathbb{E}[S]$:

$$\mathbb{E}[S] = \tilde{\lambda}_T \pi r^2 \left(\log(w) + \mathbb{E}[\log(C)] - \log\left(1 + \tilde{\lambda}_U \pi r^2\right) - \log(r) + \frac{1}{2} + O\left(\frac{1}{\tilde{\lambda}_U \pi r^2}\right)\right), \tag{44}$$

with $\mathbb{E}[\log(C)]$ as given in (7) and (8). To simplify (44), we assume $\tilde{\lambda}_U \pi r^2 > 1$. Then, using the Taylor expansion for $\log(1 + \tilde{\lambda}_U \pi r^2)$:

$$\log(1 + \tilde{\lambda}_U \pi r^2) = \log(\tilde{\lambda}_U \pi r^2) + O\left(\frac{1}{\tilde{\lambda}_U \pi r^2}\right), \tag{45}$$

resulting in (6).

## Coverage

One of the key objects of interest in cellular networks is the *coverage probability*: the number of users that can connect to at least 1 BS, which depends on the radius in which users can connect to a BS. As Fig 7 shows, the setting with optimal strength disconnects many users from the network. When the number of resources $w$ increases, the optimal radius also increases (Theorem 2), which results in a higher coverage. Therefore, in this section, we investigate the relation between the number of resources $w$ and the target coverage probability $\theta$.

The coverage probability can be analyzed with a *coverage process* [23]. The probability that a given point $\mathbf{x}$ is not covered by at least one of the circles of radius $r$ around a BS is equal to:

$$\mathbb{P}(\mathbf{x} \text{ not covered}) = \left(1 - \frac{\pi R_{\max}^2}{A}\right)^n, \tag{46}$$

where $A$ is the area of the network and $n$ the number of BSs. We assume that when $A \to \infty$, the number of BSs is asymptotically equal to $A \cdot \tilde{\lambda}_{\mathrm{T}}$:

$$\lim_{A \to \infty} \mathbb{P}(\mathbf{x} \text{ not covered}) = \lim_{A \to \infty} \left(1 - \frac{\pi R_{\max}^2}{A}\right)^{A\tilde{\lambda}_{\mathrm{T}}} = e^{-\pi\tilde{\lambda}_{\mathrm{T}} r^2}. \tag{47}$$

The probability that a given point $\mathbf{x}$ is not covered is equal to the fraction of disconnected users. Therefore, to satisfy the requirement that at least a fraction of $\theta$ of the users has a connection, $r$ needs to satisfy the inequality

$$e^{-\pi\tilde{\lambda}_{\mathrm{T}} r^2} \le 1 - \theta, \tag{48}$$

$$r \ge \sqrt{\frac{-\log(1 - \theta)}{\pi\tilde{\lambda}_{\mathrm{T}}}} =: r^{min}. \tag{49}$$

Thus, for a given coverage probability $\theta$, the radius in which a user connects to a BS should be less than or equal to $r^{min}$. To reach the target coverage, and achieve optimal strength, $r^{opt}$ has to be larger or equal to $r^{min}$, which yields

$$r^{opt} = \sqrt[3]{\frac{w}{\tilde{\lambda}_{\mathrm{U}}\pi} e^{-1+\mathbb{E}[\log(C)]}} \ge \sqrt{\frac{-\log(1 - \theta)}{\pi\tilde{\lambda}_{\mathrm{T}}}} = r_{\min}, \tag{50}$$

$$w \ge \left(\frac{-\log(1 - \theta)}{\pi\tilde{\lambda}_{\mathrm{T}}}\right)^{3/2} \frac{\tilde{\lambda}_{\mathrm{U}}\pi}{e^{-1+\mathbb{E}[\log(C)]}}. \tag{51}$$

Fig 10 shows the required bandwidth $w$ to achieve 90% coverage. The results are in line with the results on gain and expected strength: up until a threshold value of the co-location factor $p$, sharing resources is beneficial and results in less bandwidth (and therefore less costs) per base station.

Since a bandwidth of $10^7 - 10^8$ is very high in a realistic setting, there will always be a trade-off between throughput (*strength*) and coverage, depending on the cost of placing a new base station, the number of users and the number of base stations.

## Real-world scenario

We now test our results on a real-world scenario with available BS location data from Antenneregister (www.antenneregister.nl) for the province Utrecht in the Netherlands. Three

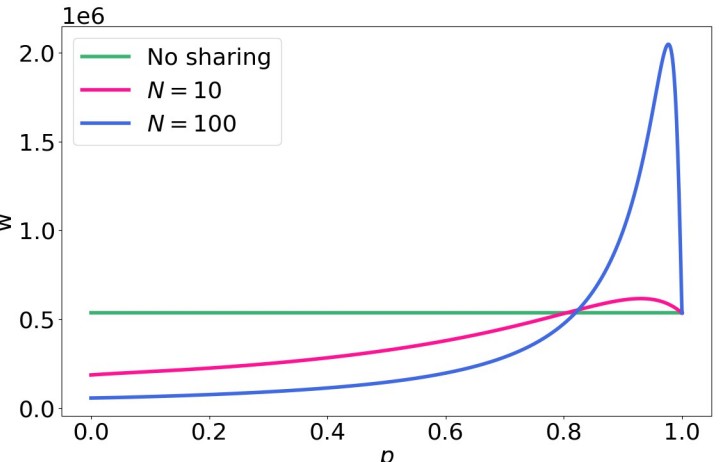

**Fig 10. Required bandwidth $w$ for different values of $p$ with a coverage target $\theta = 0.9$.**

different operators are active in this area and we analyse the network when two of these operators, called operator 1 and operator 2, share the network (Fig 11). Based on the results in [4], we took the operator with best and worse performance in the Netherlands to show how sharing benefits them both. Both operators have a slightly different BS density and we ordered them such that operator 1 has the largest density $\lambda_{BS}$ and $\lambda_{BS}^{(2)} = \frac{387}{434} \lambda_{BS}$.

We simulate users as a Poisson Point Process with density $\lambda_U = 1 \cdot 10^{-5}$ per m$^2$ in the region of the province Utrecht and assume that every BS has available bandwidth $w = 10$ MHz. Since height-differences in the BS locations can be present even when they are located on the same tower, we assume BSs are co-located when their distance is less than 5m apart. This assumption results in a co-location factor $p = 0.14$.

We calculate the strength for each link, given the optimal radius calculated in (10). When operator 1 and 2 operate alone this is 4892m for both operators, while under resource sharing, the optimal radius is 3951 m. We compare the average user strength in these real-world simulations to the expected strength calculated in (19) in Fig 12. Interestingly, even though the BSs in Fig 11 do not resemble a homogeneous PPP, the calculated strength is reasonably close to the simulated average strength. Next to the real-world scenario, we show two scenario's in which we increased the co-location factor to $p = 0.5$ and $p = 1$ by randomly choosing a BS of operator 2 and moving this BS to a location of a BS of operator 1. This experiment shows that an increased co-location factor degrades the average link strength strongly, which also follows from our theorems. In all three cases of $p$, the combined expected user strength is reasonably close to the average simulated user strength.

## Conclusion

In this paper, we show how co-location impacts the benefit of sharing resources in a wireless network using stochastic geometry. In this scenario, operators can share their base stations (BSs). In contrast to previous research, we take into account that these BSs are often located on the same tower, which strongly impacts the gains of resource sharing. We give an analytic expression for the expected *link strength* of a user by deriving the distributions of the distance to a BS, the number of resources per BS and the degree of a BS and user. Moreover, we derive the optimal radius $r$ within which BSs should allow users to connect, depending on the amount

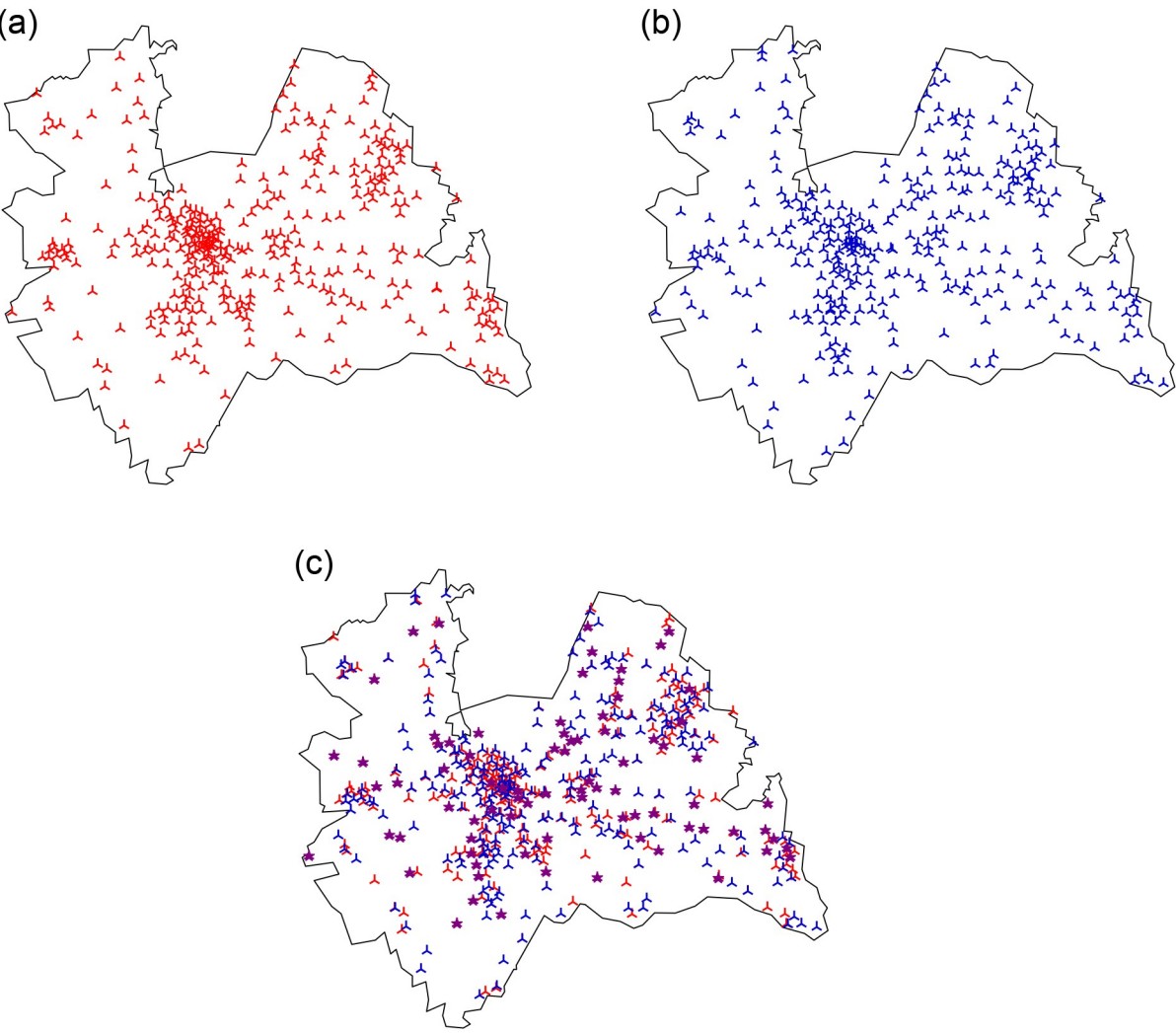

**Fig 11. BS locations of operator 1 and operator 2 in Gelderland.** (a) Operator 1: $\lambda_{BS}^{(1)} = 2.78 \cdot 10^{-7}$. (b) Operator 2: $\lambda_{BS}^{(2)} = 2.48 \cdot 10^{-7}$. (c) Operators 1 and 2 together, the purple stars denote the shared locations.

of co-located BSs. This radius decreases in the number of operators in the system. Given this optimal $r$, we then derived the sharing *gain*, defined as the optimal strength under sharing divided by the optimal strength per operator without sharing. For both small and large $N$ we find a threshold value $p$ such that for all co-location factors smaller than this $p$, sharing will benefit all operators, while for larger $p$ it might not be beneficial to share resources. In the case of unequal operator densities, we show that users subscribed to the smaller operator will benefit from sharing for all $p$, but this is not always the case for the larger operator.

There are several possible extensions of this research. First of all, we use the disk model to connect users to BSs, which can lead to a large number of connections per user. One could think of using a different connection model, such as connecting to the $k$ closest BS or connecting to only a fraction of the BSs in range. This last model will not change the conclusions much, as the expected degree of both a BS and a user will be multiplied by a fraction $q$ in this case. Moreover, as Fig 11 shows, the real-world deployment of BSs will not follow a

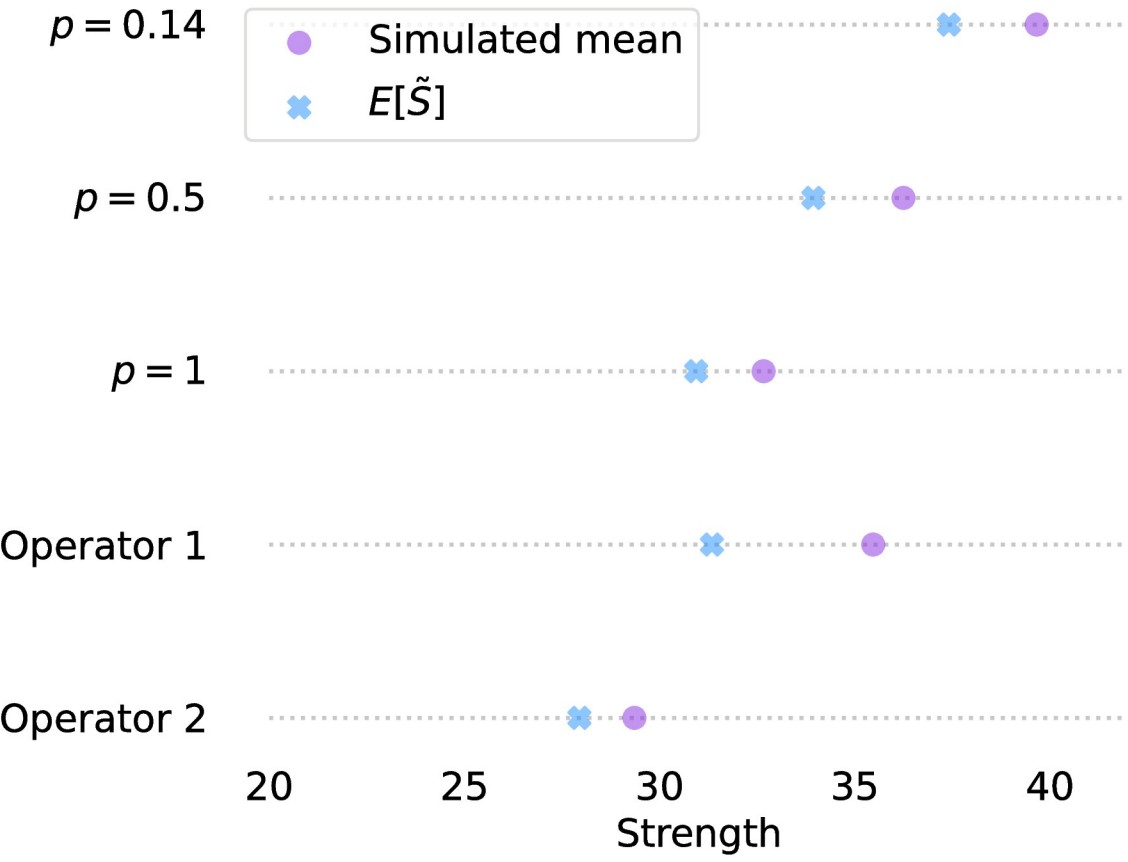

**Fig 12. Simulated and calculated strength per user for operator 1 and operator 2 separately and together.**

homogeneous PPP. Thus, extending the model to a clustered or heterogeneous PPP might make it more realistic.

Another option is to investigate partial sharing, where operators only share a fraction of their BSs with other operators, for example in regions where the coverage of other operators is low, leading to a spatial optimization problem.

Finally, while we here investigate the benefits of sharing from the perspective of link quality, there is of course also a financial aspect in sharing. When one operator shares all its resources with another operator, this makes it attractive for another operator to not expand their network, as they can benefit from other operators doing so. Therefore, it would be interesting to investigate a cost allocation model for sharing among different operators such that it still increases link quality, but also makes sharing fair, and still gives an incentive for operators is expand and improve their networks.

## Author Contributions

**Data curation:** Lotte Weedage.

**Formal analysis:** Lotte Weedage.

**Investigation:** Clara Stegehuis, Lotte Weedage.

**Methodology:** Lotte Weedage.

**Resources:** Lotte Weedage.

**Software:** Lotte Weedage.

**Supervision:** Clara Stegehuis.

**Visualization:** Lotte Weedage.

**Writing – original draft:** Lotte Weedage.

**Writing – review & editing:** Clara Stegehuis, Lotte Weedage.

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
