## [Decision Letter · Decision Letter 0]

15 Nov 2023

PONE-D-23-32266Resource and location sharing in wireless networksPLOS ONE

Dear Dr. Weedage,

Thank you for submitting your manuscript to PLOS ONE. After careful consideration, we feel that it has merit but does not fully meet PLOS ONE’s publication criteria as it currently stands. Therefore, we invite you to submit a revised version of the manuscript that addresses the points raised during the review process.

We look forward to receiving your revised manuscript.

Kind regards,

Talib Al-Ameri, Ph.D

Academic Editor

PLOS ONE

Journal Requirements:

 "C.S. is supported through NWO Veni grant 202.001."

3. Please expand the acronym “NWO Veni” (as indicated in your financial disclosure) so that it states the name of your funders in full.

"C.S. is supported through NWO Veni grant 202.001"

 "C.S. is supported through NWO Veni grant 202.001"

Reviewers' comments:

Reviewer's Responses to Questions

**Comments to the Author**

1. Is the manuscript technically sound, and do the data support the conclusions?

Reviewer #1: Yes

Reviewer #2: Partly

2. Has the statistical analysis been performed appropriately and rigorously? 

Reviewer #1: N/A

Reviewer #2: I Don't Know

3. Have the authors made all data underlying the findings in their manuscript fully available?

Reviewer #1: Yes

Reviewer #2: No

4. Is the manuscript presented in an intelligible fashion and written in standard English?

Reviewer #1: Yes

Reviewer #2: Yes

5. Review Comments to the Author

Reviewer #1: The paper titled "Resource and location sharing in wireless networks" is presented well; however, i have some suggestion for improvement, which are:

1. Explain the problem statement in the introduction section clearly.

2. Each section needs proper label numbering, or label it with a bullet-point section heading and subheading number.

3. Review and cite some of the latest references in the literature.

a. Ao, S., Niu, Y., Han, Z., Ai, B., Zhong, Z., Wang, N., & Qiao, Y. (2023). Resource Allocation for RIS-Assisted Device-to-Device Communications in Heterogeneous Cellular Networks. IEEE Transactions on Vehicular Technology.

b. Khan, M. N., Rahman, H. U., Khan, M. Z., Mehmood, G., Sulaiman, A., Shaikh, A., & Alqhatani, A. (2022). Energy-efficient dynamic and adaptive state-based scheduling (EDASS) scheme for wireless sensor networks. IEEE Sensors Journal, 22(12), 12386–12403.

c. Haq, M. Z. U., Khan, M. Z., Rehman, H. U., Mehmood, G., Binmahfoudh, A., Krichen, M., & Alroobaea, R. (2022). An adaptive topology management scheme to maintain network connectivity in Wireless Sensor Networks. Sensors, 22(8), 2855.

4. The values of the graph in Fig 6. is not clear; please explain why such output is produced.

5. Is the real-time scenario much like one of the real-time solutions?

6. Cite the figures that you have taken from other papers.

8. There are many typographical and grammatical mistakes that need correction.

Reviewer #2: This paper studied the resource sharing problem in the context of base station (BS) co-location in wireless networks with multiple service operators. More specifically, assuming a given co-location overlapping factor, the authors define a new metric called “link strength” and derived the expected value of the link strength as a function of the overlapping factor and found the optimal overlapping factor. The paper is well written. However, I have the following concerns regarding the technical contributions.

1. The Poisson point process (PPP) is used to model the stochastic distribution of the BSs and users. In PPP, there is no minimum distance separation, i.e., points can be arbitrarily close to each other. This is not true in practice—the is usually a minimum distance between BSs (say at least 200 meters apart from each other). This minimum distance can have significant impact on the network capacity analysis. It would be better to consider such more practical models (I am not sure if this is doable mathematically).

2. Page 4, line 114, the overlapping factor p has not been rigorously defined.

3. Page 5, line 120, what is the “degree” of a BS? Is it the number of users that are connected to this BS?

4. Regarding the channel model (eq. (4)), it seemed that users of the same BS are allocate nonoverlapping sub-bands so they do not interfere with each other. Therefore, SNR instead of SINR is used. The channel model should be clearly described. In practice, inter-BS interference exists which seems to be omitted in the channel model. How do you justify this simplification?

5. What is the rationale behind the definition of the utility metric “user strength”? To enforce fairness among users (i.e., users with low throughput should be prioritized a little bit such they can get adequate service), a concave utility function (say logarithm function) is typically used on top of the Shannon throughput. However, the utility definition is a bit counterintuitive as it takes the SNR out of the logarithm of Shannon capacity. How do you justify the throughput performance under this utility function?

6. In Fig. 3, the authors claimed that the average channel capacity is a decreasing function of r, then why is the first point with a very small r having a very low capacity? In addition, the power decay factor alpha is set to be 1 which is not practical at all—it is known that in a free space radiation model, the power decays at least quadratically with the distance. In practical scenarios alpha can only be larger. It seemed that in eq (5) it is also assumed a decay factor of 1.

7. An upper bound on p has been derived in eq (18) and a comparison between this bound and the true values is shown in Fig. 6. In Fig. 6, for a given beta_2, why is that the upper bound is smaller than the true values?

Minor issues and typos:

1. In Fig. 6, part of the x-axis label is cut off which should be corrected.

6. PLOS authors have the option to publish the peer review history of their article (what does this mean?). If published, this will include your full peer review and any attached files.

Reviewer #1: No

Reviewer #2: No

---

## [Author Response · Author response to Decision Letter 0]

8 Jan 2024

Dear editor and reviewers,

We thank you for your comments and revised the manuscript accordingly. Our responses are in the attached pdf.

Kind regards,

Lotte Weedage

---

## [Decision Letter · Decision Letter 1]

9 Feb 2024

Resource and location sharing in wireless networks

PONE-D-23-32266R1

Dear Dr. Weedage,

We’re pleased to inform you that your manuscript has been judged scientifically suitable for publication and will be formally accepted for publication once it meets all outstanding technical requirements.

Kind regards,

Talib Al-Ameri, Ph.D

Academic Editor

PLOS ONE

Reviewers' comments:

Reviewer's Responses to Questions

**Comments to the Author**

1. If the authors have adequately addressed your comments raised in a previous round of review and you feel that this manuscript is now acceptable for publication, you may indicate that here to bypass the “Comments to the Author” section, enter your conflict of interest statement in the “Confidential to Editor” section, and submit your "Accept" recommendation.

Reviewer #2: All comments have been addressed

2. Is the manuscript technically sound, and do the data support the conclusions?

Reviewer #2: Yes

3. Has the statistical analysis been performed appropriately and rigorously? 

Reviewer #2: Yes

4. Have the authors made all data underlying the findings in their manuscript fully available?

Reviewer #2: Yes

5. Is the manuscript presented in an intelligible fashion and written in standard English?

Reviewer #2: Yes

6. Review Comments to the Author

Reviewer #2: My comments and concerns regarding the technical and presentation in the previous revision have been addressed.

7. PLOS authors have the option to publish the peer review history of their article (what does this mean?). If published, this will include your full peer review and any attached files.

Reviewer #2: No

---

## [Editor Report · Acceptance letter]

13 Feb 2024

PONE-D-23-32266R1 

PLOS ONE

Dear Dr. Weedage, 

I'm pleased to inform you that your manuscript has been deemed suitable for publication in PLOS ONE. Congratulations! Your manuscript is now being handed over to our production team.

Kind regards, 

on behalf of

Dr. Talib Al-Ameri 

Academic Editor

PLOS ONE